# Enhancing Microstructural and Mechanical Properties of Ferrous Medium-Entropy Alloy through Cu Addition and Post-Weld Heat Treatment in Gas Tungsten Arc Welding

**DOI:** 10.3390/ma17010181

**Published:** 2023-12-28

**Authors:** Seonghoon Yoo, Yoona Lee, Myeonghawn Choi, Hyunbin Nam, Sangyong Nam, Namhyun Kang

**Affiliations:** 1Department of Materials Science and Engineering, Pusan National University, Busan 46241, Republic of Korea; tjdgns2198@pusan.ac.kr (S.Y.); yuna584979@gmail.com (Y.L.); enejwl3271@naver.com (M.C.); 2Department of Joining Technology, Korea Institute of Materials Science, Changwon 51508, Republic of Korea; hbnam12@kims.re.kr; 3Department of Materials Engineering and Convergence Technology, Green Energy Convergence Research Institute, Gyeongsang National University, Jinju 52828, Republic of Korea; walden@gnu.ac.kr

**Keywords:** high-entropy alloy, metals and alloys, design of materials, gas tungsten arc welding, post-weld heat treatment, recrystallisation, heat-affected zone

## Abstract

This study investigates the impact of a high-entropy alloy filler metal coated with copper (Cu) and post-weld heat treatment (PWHT) on the weldability of a ferrous medium-entropy alloy (MEA) in gas tungsten arc welding. The addition of 1-at% Cu had an insignificant effect on the microstructural behaviour, despite a positive mixing enthalpy with other elements. It was observed that a small amount of Cu was insufficient to induce phase separation into the Cu-rich phase and refine the microstructure of the as-welded specimen. However, with an increase in the PWHT temperature, the tensile strength remained mostly consistent, while the elongation significantly increased (elongation of as welded, PWHT700, PWHT800, and PWHT 900 were 19, 43, 55 and 68%, respectively). Notably, the PWHT temperature of 900 °C yielded the most desirable results by shifting the fracture location from the coarse-grained heat-affected zone (CGHAZ) to base metal (BM). This was due to significant recrystallisation and homogenised hardness of the cold-rolled BM during PWHT. However, the CGHAZ with coarse grains induced by the welding heat input remained invariant during the PWHT. This study proposes a viable PHWT temperature (900 °C) for enhancing the weldability of cold-rolled ferrous MEA without additional process.

## 1. Introduction

High-entropy alloy (HEA) exhibits excellent phase stability and mechanical properties attributed to its four core effects: high entropy, lattice distortion, sluggish diffusion, and cocktail effect [1,2,3,4]. These characteristics simplify the alloy design process for HEA and enhance its potential for widespread commercial use in various industries [5,6,7,8,9]. For use as a structural material, the HEA should be assembled by a welding process, necessitating the development of a new welding material. Numerous studies explored the replacement of face-centred cubic (FCC)-based structural steels with HEA to improve weldability [10,11,12]. In structural materials, the microstructure coarsened by welding heat input severely deteriorates the mechanical properties, becoming the primary reason for degradation in weldability [13,14,15,16,17]. Therefore, this weldability evaluation aims to determine a solution for preventing fractures in the weld zone (WZ) and coarse-grained heat-affected zone (CGHAZ).

There were numerous studies to improve the performance of weldments by enhancing and stabilising the microstructure and mechanical properties in the WZ and/or CGHAZ. Bae et al. [18] designed a ferrous medium-entropy alloy (MEA) of Fe_60_Co_15_Ni_15_Cr_10_, featuring an FCC crystal structure. The cobalt (Co) and nickel (Ni) contents were reduced compared to the Cantor HEA (CoCrFeMnNi), resulting in improved tensile properties attributed to its metastable FCC phase. However, the presence of a small amount of body-centred cubic (BCC) in its microstructure could potentially lead to poor mechanical properties in the weld joints [19,20,21]. To address this issue, the utilisation of a HEA filler metal (Cantor alloy) was considered for stabilising austenite and suppressing the formation of BCC in the WZ [22]. Furthermore, the introduction of copper (Cu), with a positive mixing enthalpy, induced phase separation from the primary FCC, leading to the formation of a Cu-rich FCC crystal structure [23,24,25]. Several studies have highlighted the beneficial effects of a Cu-rich phase in delaying atomic diffusion and producing a fine microstructure, thereby improving the mechanical properties [23]. Moreover, the Cu-rich phase exhibits higher Vickers hardness than the primary FCC phase, hindering dislocation movement [26,27]. These phenomena have a strengthening effect, improving the overall performance of the weldments.

Weldments must meet specific criteria to ensure capable performance [28]. Nam et al. [29] reported that CrMo-based inclusions existed in the interdendritic area after laser beam welding of a Cantor alloy. However, PWHT notably reduced the size and volume fraction of inclusions in the WZ, leading to a substantial increase in ductility. Chen et al. [30] reported that increasing PWHT temperature (from 650 to 850 °C) showed homogenised microstructure and shifting of fracture location from fusion line including Ni-based inter-metallic compound (IMC) to the NiTi BM. It was owing to the thermally activated system by PWHT that promoted the Ni_3_Ti equilibrium phase in the fusion line. Additionally, PWHT can also relieve the residual stress induced by welding heat input [31]. This means the PWHT process has a remarkable influence on the weldment performance. Therefore, it is necessary to study the effect of PWHT on the microstructure and mechanical properties of new welding materials.

This study evaluated the weldability of metastable MEA to design the filler metal composition and the PWHT process. The investigation included microstructural evolution, such as the phase separation of the Cu-rich phase and recrystallisation behaviour after welding and PWHT. In addition, the fracture location with respect to PWHT temperature was analysed based on the correlation between the microstructure and mechanical properties. The objective of the study is to suggest a viable filler metal design and PWHT temperature to enhance the weldability of ferrous MEA using HEA.

## 2. Materials and Methods

### 2.1. Materials Preparation

The BM used in this study was Fe_60_Co_15_Ni_15_Cr_10_ (a ferrous MEA), which was produced by vacuum induction melting. An ingot was homogenised at 1100 °C for 24 h, followed by hot rolling from 145 to 3 mm in thickness, air cooling, and subsequent cold rolling from 3 to 1.5 mm thick at room temperature. Figure 1 depicts the schematic diagram of the single-pass gas tungsten arc (GTA) welding process applied to a V groove (angle of 60°) with a filler metal. We optimised the welding conditions that ensured no macroscopic defects and achieved full penetration in the welding process. Therefore, this study used welding conditions such as a current of 90 A, voltage of 12.5 V, weld speed of 22 cm/min, and heat input of 3.1 kJ/cm. The filler metal was a Cantor-based HEA coated with 1 at % Cu, that is, (CrCo-FeMnNi)99Cu1. Subsequently, PWHT was conducted at 700, 800, and 900 °C for 1 h, considering the recrystallisation temperature for the FCC structure, followed by quenching (named PWHT700, PWHT800, and PWHT900, respectively).

### 2.2. Experiment Methods

Microstructural and mechanical analyses were conducted to investigate the weldability behaviour post welding and PWHT. The tensile fracture location was observed through optical microscopy after etching in a solution of 100 mL of ethanol + 4 g picric acid + 5 mL hydrochloric acid. X-ray diffraction (XRD) was employed to identify the constituent phases in the WZ. The θ/2θ scan was performed for 40–80° using Cu-Kα characteristic X-ray, with a scan width and speed of 0.05° and 1°/min, respectively. Furthermore, electron probe microanalysis (EPMA) was performed to quantify the components and analyse their segregation behaviour in the WZ. The recrystallisation behaviour with respect to the PWHT temperature was investigated using electron backscatter diffraction (EBSD). The step size of 0.5 μm was adopted for the analysis of grain orientation spread as the average grain size of the fully recrystallised BM was 23 μm [32,33]. The Vickers hardness was measured in the welds using a load of 300 gf (2.9 N) with a holding time of 10 s. Tensile testing was performed on sub-sized specimens following ASTM–E8 at room temperature with a strain rate of 0.03 mm/s. The average value and standard deviation of the tensile property were calculated from three independent tests.

## 3. Results

### 3.1. Effect of 1 at % Cu on the Microstructural Behaviour in the Weld Metal

Figure 2 represents the quantitative behaviour of the elements along the centreline in the thickness for various PWHT temperatures. The red dotted lines indicate the fusion lines. The filler metal had a higher concentration of nickel (Ni), manganese (Mn), Co, chromium (Cr), and Cu and a lower iron (Fe) content than the BM. Due to dilution, therefore, the concentrations of Ni, Mn, Co, Cr, and Cu increased and that of Fe decreased in the WZ compared to those in the BM. A quantity of 1-at% Cu was properly mixed in the WZ (Figure 2f). No macro-diffusional behaviour of the component was observed concerning the PWHT temperature, indicating that the PWHT in the range of 700–900 °C was insufficient to significantly diffuse substitutional atoms.

Figure 3 illustrates the phase identification using XRD in the WZ at various PWHT temperatures. The WZ exhibited only the FCC crystal structure, regardless of the PWHT temperature. Typically, the Cu-rich FCC phase, having the same crystal structure but a different lattice parameter compared to the primary FCC, would result in a double peak in the XRD pattern (that is, separation into secondary FCC). In addition, despite the positive mixing enthalpy of Cu and the anticipated clarity of the double peak with increasing PWHT temperature, no distinct double peaks were observed in the WZ, even at a PWHT temperature of 900 °C [9,23,26].

Figure 4 depicts the elemental maps of WZ at various PWHT temperatures, with the WZ exhibiting a dendritic microstructure. The interdendritic region was mainly segregated with Ni, Mn, and Cu, which have relatively lower melting temperatures than Fe and Co. Therefore, the interdendritic segregation region was determined to be the final solidification region (referred to as the Cu-rich region) rather than the Cu-rich phase. The content in the Cu-rich region was 2.0 ± 0.1, 2.7 ± 0.2, 2.2 ± 0.1, and 1.9 ± 0.1-at% Cu for as welded, PWHT700, PWHT800, and PWHT900, respectively. Theoretically, Cu possesses a positive mixing enthalpy with other elements (Fe, Ni, Mn, Co, and Cr) and tends to segregate from the primary phase in the thermally activated system [23]. In other words, an increase in PWHT temperature would result in a higher content of Cu in the Cu-rich region. However, the Cu content in the Cu-rich region was mostly consistent with increasing PWHT temperature. Through the EPMA analysis (Figure 3 and Figure 4), no clear Cu-rich phase (phase separation) was formed after PWHT. No phase separation in the interdendritic region was confirmed in the WZ, even though the 1-at% Cu in the filler metal was well mixed with the weld metal.

### 3.2. Effect of PWHT Temperature on Mechanical Properties of Transverse Welds

Figure 5 illustrates the hardness distribution in the transverse weld at various PWHT temperatures. As the BM was a cold-rolled ferrous MEA, the BM and CGHAZ of the as-welded specimens exhibited the highest and lowest hardness values, respectively. The CGHAZ was the most deteriorated region, where the grains grew because of the welding heat input [34,35]. The WZ and CGHAZ exhibited a consistent tendency for hardness regardless of the PWHT since they retained an as-welded microstructure without cold deformation. PWHT decreased the hardness of the BM by the annealing effect compared to that of the WZ and CGHAZ after PWHT. Furthermore, PWHT900 demonstrated the homogenised hardness in the CGHAZ and BM.

Figure 6 presents the stress–strain curves with respect to the PWHT temperatures. The average values of tensile strength for as welded, PWHT700, PWHT800, and PWHT900 were 455 ± 10.8, 450 ± 8.5, 424 ± 12.5, and 414 ± 13.8 MPa, respectively. The elongations of as welded, PWHT700, PWHT800, and PWHT900 were 19.3 ± 0.5, 43.3 ± 1.9, 54.9 ± 3.6, and 68.0 ± 0.8%, respectively. The tensile strength exhibited minimal variation, while the elongation significantly increased with increasing PWHT temperature.

Figure 7 illustrates the morphology of the transverse and the fracture locations of each specimen after tensile testing. The red dotted lines indicate the fusion lines. The as welded, PWHT700 and PWHT800 alloys (Figure 7a, Figure 7b, and Figure 7c, respectively) exhibited fractures in the CGHAZ. However, in the case of PWHT900 (Figure 7d), the fracture occurred in the BM instead of the CGHAZ. PWHT900 demonstrated the best weldability of the transverse welds, characterised by large elongation, nearly the same tensile strength, and the prevention of fractures at the WZ and CGHAZ.

To investigate the reason for the change in the tensile-fracture location, EBSD analysis in the CGHAZ and BM for the as welded, PWHT800, and PWHT900 specimens was conducted before tensile testing, as showcased in Figure 8. The red lines in Figure 8a–c indicate the fusion line where columnar grains occurred in the WZ and equiaxed grains remained in the CGHAZ. The CGHAZ near the fusion line exhibited significant grain growth produced by the welding heat input as compared to the small grain of the BM (~23 μm), correlating with the hardness drop in the CGHAZ (Figure 5). Therefore, it matches well with the CGHAZ fracture for the as welded, PWHT700, and PWHT800 specimens. Figure 8d–f indicate that the BM of the as welded, PWHT800, and PWHT900 samples were deformed, partially recrystallised, and fully recrystallised microstructures, respectively. The fully recrystallised microstructure of the BM and the homogenised hardness values between the CGHAZ and BM (Figure 5) of PWHT900 were associated with BM fracture during tensile testing.

## 4. Discussion

### 4.1. Mixing Enthalpy of Primary FCC and Cu-Rich Region

The addition of 1-at% Cu produced insignificant phase separation into the Cu-rich phase in the WZ of the ferrous MEA (Figure 3 and Figure 4). In the regular solution of a binary system, the mixing enthalpy of positive and negative values produces a cluster and solid solution, respectively [23]. Table 1 lists the mixing enthalpies of the binary systems [36]. Cu has a positive mixing enthalpy with all other elements [37]. In addition, as the positive mixing enthalpy increased, the tendency to separate from the primary FCC to the secondary FCC increased.

The mixing enthalpy of a multicomponent system can be calculated assuming that it is a binary system. Equations (1) and (2) show the calculations for the mixing enthalpy of a multicomponent system [3]:(1)$$\Delta H_{mix}=\sum_{i=1}^{n}\Omega_{ij}c_{i}c_{j}$$
(2)$$\Omega_{ij}=4{\Delta H}_{mix}^{AB}$$
where Ω denotes the mixing enthalpy of binary *i*/*j* pair and $c_{i}$ represents the concentration of the individual element. Table 2 lists the mixing enthalpies calculated using the concentrations measured in the primary FCC and Cu-rich regions in the WZ for various PWHT temperatures. The slight increase in the mixing enthalpy from the primary FCC to the Cu-rich region due to the segregation of Cu suggested an insignificant variation in stabilising the secondary phase thermodynamically for the as-welded specimen. Furthermore, an increase in PWHT temperature produced similar mixing enthalpy values for the primary FCC and Cu-rich regions. Therefore, the amount of 1-at% Cu in the filler metal was insufficient to produce the Cu-rich phase, even though thermal activation was applied by PWHT; this result was congruent with the XRD and EPMA observations (Figure 3 and Figure 5).

### 4.2. Variation of the Driving Force for Recrystallisation during the PWHT

The as-welded, PWHT700, and PWHT800 specimens exhibited fractures near the CGHAZ. However, PWHT900 underwent fracture at the BM despite having a CGHAZ. Figure 9 illustrates the inverse pole figure (IPF) map and corresponding misorientation profiles for the CGHAZ and BM in the PWHT900 specimen. The red line in Figure 9a represents the fusion line, while black lines within the grain (Figure 9a,b) indicate the lines used to analyse the misorientation profile. Notably, Figure 9c illustrates that the misorientation of some CGHAZ grains continuously increases concerning the distance from the origin. This suggests that the coarse grains of the CGHAZ experienced a low driving force for recrystallisation by the PWHT, retaining incompletely recrystallised grains with substructures and dislocations. In other words, PWHT produced insignificant softening of the CGHAZ, and PWHT700, PWHT800, and PWHT900 demonstrated the same hardness as the CGHAZ, as shown in Figure 5. However, the BM exhibited minimal variation in misorientation within a grain. The misorientation indirectly represents the density of dislocation, and the BM is presumed to have a softened microstructure due to the full recrystallisation induced by high PWHT temperature (900 °C) [38]. Given that the BM in the as-welded specimen was deformed by cold rolling (Figure 8d), the PWHT induced a larger driving force for recrystallisation than the CGHAZ. Consequently, the hardness of BM and CGHAZ were homogenised and the fracture occurred in the BM of PWHT900. The PWHT temperature of 900 °C shifted the fracture location from the CGHAZ to the BM. PWHT900 was found to be the most efficient temperature for enhancing the weldability of the cold-rolled ferrous MEA.

## 5. Conclusions

This study investigated the weldability of a ferrous MEA with a Cu-coated HEA filler metal to propose a viable PWHT temperature. To explain the enhanced weldability by shifting fracture position in the highest PWHT temperature, the EBSD and hardness analysis was conducted across the transverse weld to understand the recrystallisation behaviour. The significant findings of this study are as follows:(1)Cu content of 1 at% was observed in the WZ, resulting in sound welds with the filler metal properly mixed in the WZ.(2)The WZ comprises only an FCC crystal structure. After the PWHT (700–900 °C), the WZ exhibited primary FCC without secondary FCC (Cu-rich phase) expected due to the positive mixing enthalpy of Cu. A small amount of Cu (1 at. %) in the WZ produced a segregated region in the final interdendritic region (Cu-rich region) with no Cu-rich phase. Therefore, the addition of 1-at% Cu was insufficient to produce phase separation into the Cu-rich region, even though thermal activation was applied by PWHT.(3)For the as-welded specimen, the BM demonstrated the highest hardness, attributed to the cold-rolled and deformed microstructures. The CGHAZ had the lowest hardness (decrease in hardness) due to grain growth, regardless of the PWHT temperature. An increase in the PWHT temperature significantly decreased the hardness of the BM by recrystallisation, and PWHT900 produced the homogenised hardness as the CGHAZ and BM.(4)As the PWHT temperature increased, the tensile strength displayed almost the same value, and the elongation increased significantly. The fracture locations of the as-welded, PWHT700, and PWHT800 specimens were located in the CGHAZ, where the lowest hardness values were observed. However, PWHT900 fractured at the BM, which had the homogenised hardness as the CGHAZ.(5)The CGHAZ of PWHT900 exhibited substructures and dislocations in the grains. The CGHAZ had a low driving force for recrystallisation during the PWHT. The cold-rolled BM displayed higher recrystallisation driving forces than the CGHAZ. Therefore, the BM and CGHAZ of PWHT900 had homogenised hardness during PWHT. The variation in the driving force homogenised the hardness of the BM and shifted the fracture location from the CGHAZ to the BM in the PWHT900.

## Figures and Tables

**Figure 1 materials-17-00181-f001:**
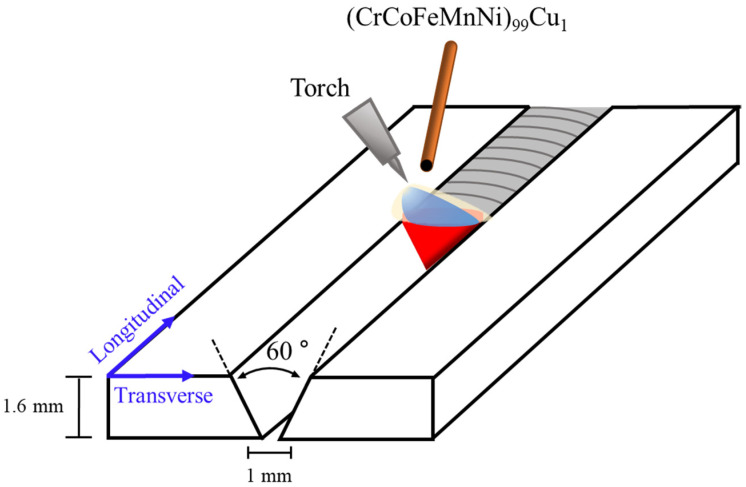
Schematic diagram of GTA welding process using Cu-coated HEA filler metal.

**Figure 2 materials-17-00181-f002:**
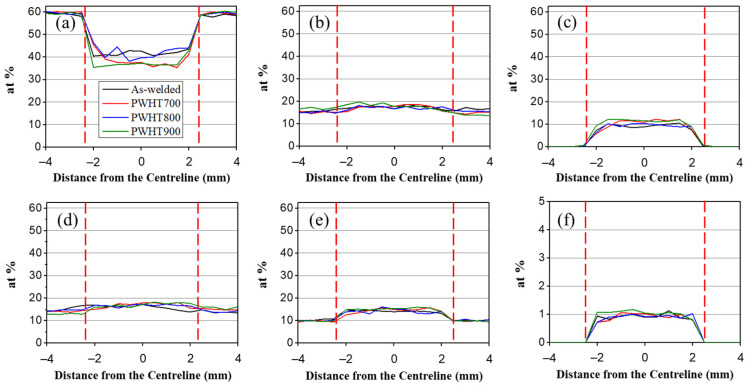
Quantitative EPMA in weld metal for various PWHT temperatures and elements: (**a**) Fe; (**b**) Ni; (**c**) Mn; (**d**) Co; (**e**) Cr; and (**f**) Cu.

**Figure 3 materials-17-00181-f003:**
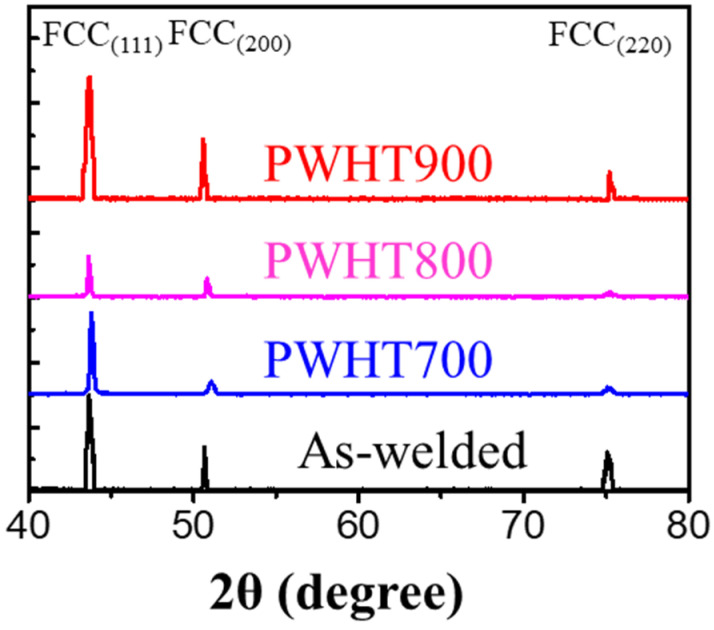
XRD of WZ for various PWHT temperatures.

**Figure 4 materials-17-00181-f004:**
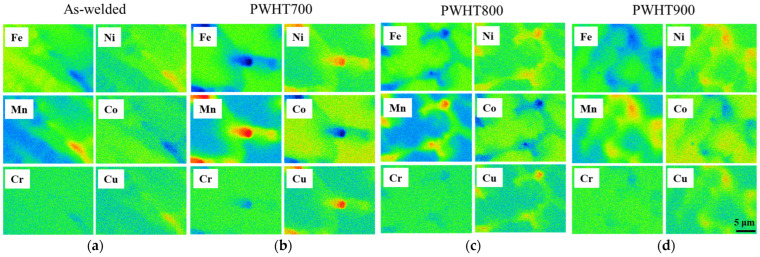
Elemental maps determined by EPMA in WZ for various PWHT temperatures: (**a**) as welded; (**b**) 700 °C; (**c**) 800 °C; and (**d**) 900 °C.

**Figure 5 materials-17-00181-f005:**
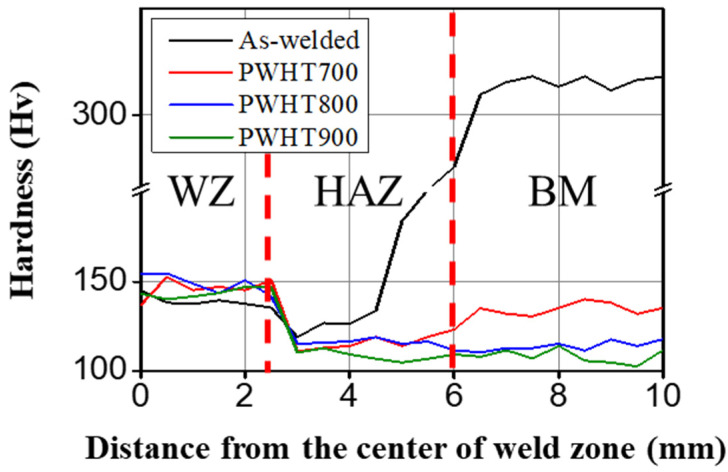
Hardness distribution of transverse welds for various PWHT temperatures.

**Figure 6 materials-17-00181-f006:**
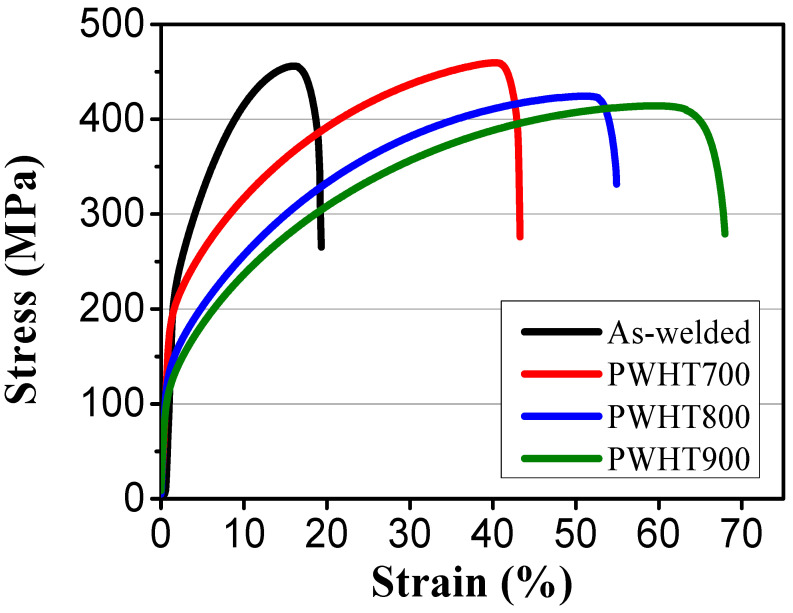
Tensile property of the transverse welds followed by various PWHT temperatures.

**Figure 7 materials-17-00181-f007:**
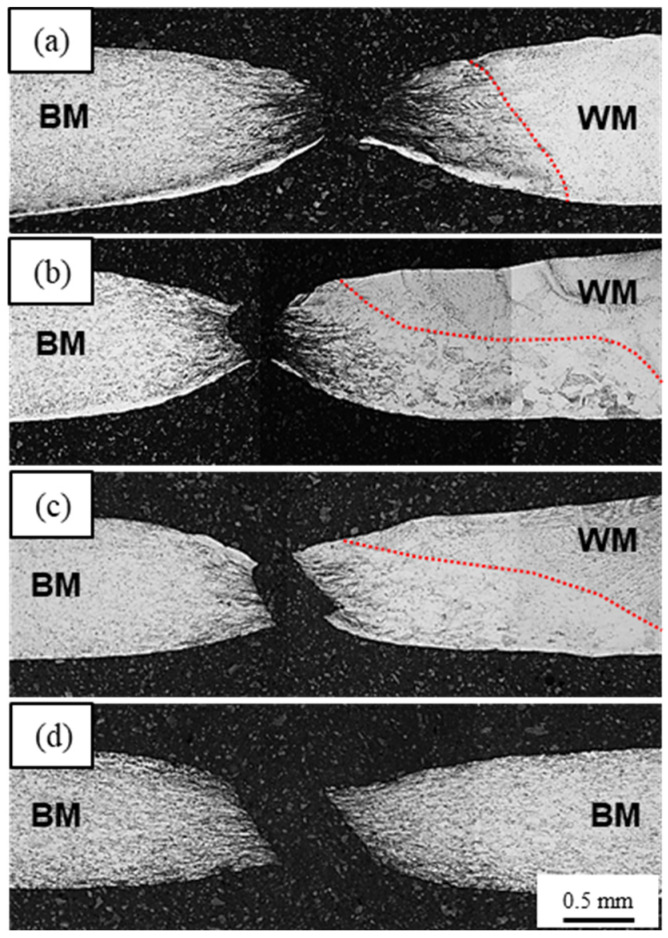
Fracture location of transverse welds produced by various PWHT temperatures: (**a**) as welded; (**b**) 700 °C; (**c**) 800 °C; and (**d**) 900 °C.

**Figure 8 materials-17-00181-f008:**
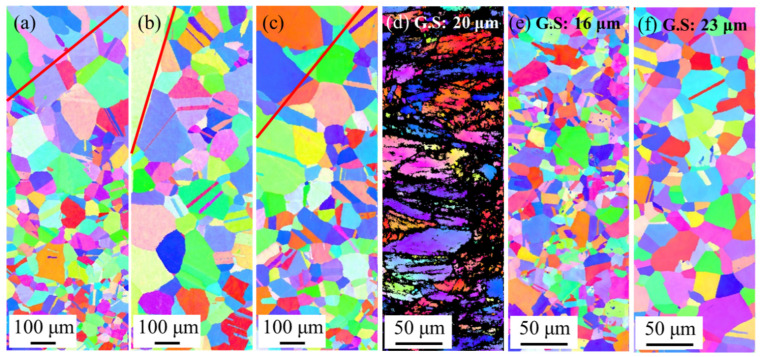
EBSD analysis with inverse pole figure (IPF) map in (**a**–**c**) CGHAZ and (**d**–**f**) BM with respect to the PWHT temperature: (**a**,**d**) as welded; (**b**,**e**) 800 °C; and (**c**,**f**) 900 °C.

**Figure 9 materials-17-00181-f009:**
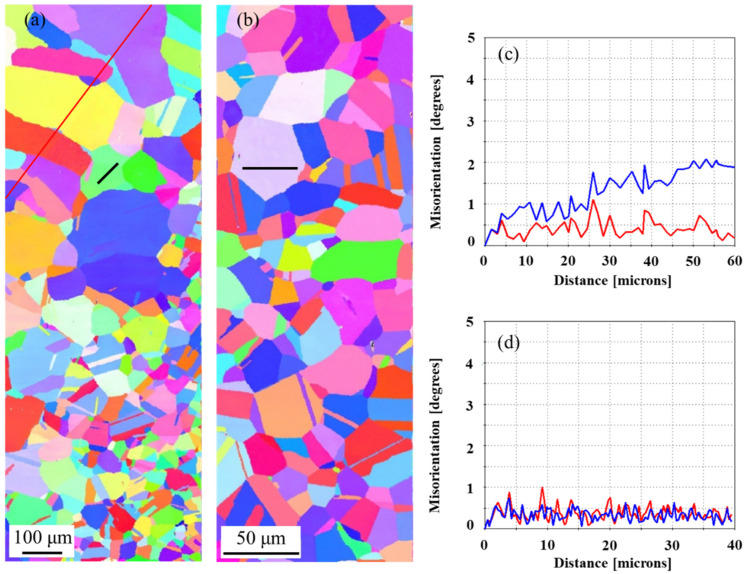
IPF map and corresponding misorientation profiles for the PWHT900 measured in various locations: (**a**,**c**) CGHAZ and (**b**,**d**) BM. (Red lines mean point to point and blue lines mean point to origin).

**Table 1 materials-17-00181-t001:** Mixing enthalpy of various elements in the binary system of MEA.

	Co	Cr	Fe	Mn	Ni	Cu
Co	-	–4	–4	5	0	6
Cr	-	-	–1	2	–7	12
Fe	-	-	-	0	–2	13
Mn	-	-	-	-	–8	4
Ni	-	-	-	-	-	4
Cu	-	-	-	-	-	-

**Table 2 materials-17-00181-t002:** Mixing enthalpy calculated for primary FCC and Cu-rich region in the WZ for various PWHT temperatures.

	Zone	$\Delta H_{mix}$ [kJ/mol]
As welded	Primary FCC	–2.7
	Cu-rich region	–2.5
PWHT700	Primary FCC	–2.8
	Cu-rich region	–2.6
PWHT800	Primary FCC	–2.8
	Cu-rich region	–2.6
PWHT900	Primary FCC	–2.8
	Cu-rich region	–2.5

## Data Availability

The data presented in this study are available upon request from the corresponding author.

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
