# Peer review of "Enhancing Microstructural and Mechanical Properties of Ferrous Medium-Entropy Alloy through Cu Addition and Post-Weld Heat Treatment in Gas Tungsten Arc Welding"

_materials, 2023, doi:10.3390/ma17010181_

Round 1

Reviewer 1 Report

Comments and Suggestions for Authors

This study investigates the impact of a high-entropy alloy filler metal coated with copper 14 (Cu) and post-weld heat treatment (PWHT) on the weldability of a ferrous medium-entropy alloy 15 (MEA) in gas-tungsten arc welding. The work is very interesting and can be published in this journal after minor revision:

(1) the font size in figure 3 is too big.

(2) sevaral important work should be mensioned:

Shi, J., Ling, S., Kuang, Y., Tong, Y., Hu, Y. and Deng, D. (2023), "Influence of microstructure of CoCrNi medium entropy alloy on its corrosion behavior", Anti-Corrosion Methods and Materials, Vol. 70 No. 6, pp. 438-448. https://doi.org/10.1108/ACMM-06-2023-2840

(3) Figure 7, does the author observe the cross section of the sample? The mophology of the cross-section give important information about the frature mechanism.

Author Response

The authors are really appreciate for taking the time to review this manuscript. Please see the attachment.

Reviewer 2 Report

Comments and Suggestions for Authors

This study presents the impact of a high-entropy alloy filler metal coated with copper (Cu) and post-weld heat treatment (PWHT) on the weldability of a ferrous medium-entropy alloy (MEA) in gas-tungsten arc welding. The title of the article is new and practical and attractive to the reader. The following should also be considered before acceptance.

The abstract can be made more attractive by using more quantitative data. It is suggested to provide more quantitative results.

The manuscript needs general writing and grammar editing.

The way of referencing in the introduction should be modified. The use of general sentences with more than four references can be seen in the introduction (Lines 33, and 40 ).

The first paragraph of introduction presented are primarily general and general information. At the end of the introduction, a suitable summary of the importance of the present issue should be provided.  Also, discontinuity between paragraphs is evident in most of the introduction. It is suggested to rewrite the introduction.

Use the following resources to deepen the introduction and discussion. Effects of post-weld heat treatment on the microstructure and mechanical properties of laser-welded NiTi/304SS joint with Ni filler. Mechanism and technology evaluation of a novel alternating-arc-based directed energy deposition method through polarity-switching self-adaptive shunt. Microstructure and mechanical properties of ultrasonic spot welding TiNi/Ti6Al4V dissimilar materials using pure Al coating.

The process of selecting laser welding parameters should be presented. Explain how to choose optimum welding parameters. How is the welding quality checked? Explain more about the welding process.

How has the reproducibility of these results been checked? Specify the number of tensile tests conducted. Error bar added to mechanical properties results.

How are the results of Figure 2 extracted? Mention the accuracy of the measurement.

The paragraph related to Figure 4 (elemental analysis) should be rewritten. This part is not very clear.

The results section is well organized and categorized. But some parts report the results, which require corrections and deepening the analysis and discussion.

The conclusion needs rewriting. In the conclusion section, a summary of the purpose of the research, innovation, and research method should be presented before presenting the highlights.

Comments on the Quality of English Language

***

Author Response

(The authors gave the same response as above.)

Reviewer 3 Report

Comments and Suggestions for Authors

The paper presents a study of a medium entropy alloy weld in a Cantor alloy base material, and the effect of PWHT. The paper is fine, and I only have two comments.

1. As you only have 1% Cu, the volume fraction of fcc-Cu would be a maximum of 1 volume-%. Do you really think this would be detectable using XRD?

2. In the EPMA maps, there is no microbial.

Comments on the Quality of English Language

The English is easy to understand, but the language could be improved (things like a/an, the).

Author Response

The authors are really appreciate for taking time to review this manuscript. Please see the attachment.
